# The Contribution of Viral Proteins to the Synergy of Influenza and Bacterial Co-Infection

**DOI:** 10.3390/v14051064

**Published:** 2022-05-16

**Authors:** Miriam Mikušová, Karolína Tomčíková, Katarína Briestenská, František Kostolanský, Eva Varečková

**Affiliations:** Biomedical Research Center of the Slovak Academy of Sciences, Institute of Virology, Dúbravská Cesta 9, 845 05 Bratislava, Slovakia; miriam.mikusova@savba.sk (M.M.); karolina.tomcikova@savba.sk (K.T.); katarina.briestenska@savba.sk (K.B.); virufkos@savba.sk (F.K.)

**Keywords:** influenza virus, influenza viral proteins, *Streptococcus pneumoniae*, co-infection, secondary bacterial infection

## Abstract

A severe course of acute respiratory disease caused by influenza A virus (IAV) infection is often linked with subsequent bacterial superinfection, which is difficult to cure. Thus, synergistic influenza–bacterial co-infection represents a serious medical problem. The pathogenic changes in the infected host are accelerated as a consequence of IAV infection, reflecting its impact on the host immune response. IAV infection triggers a complex process linked with the blocking of innate and adaptive immune mechanisms required for effective antiviral defense. Such disbalance of the immune system allows for easier initiation of bacterial superinfection. Therefore, many new studies have emerged that aim to explain why viral–bacterial co-infection can lead to severe respiratory disease with possible fatal outcomes. In this review, we discuss the key role of several IAV proteins—namely, PB1-F2, hemagglutinin (HA), neuraminidase (NA), and NS1—known to play a role in modulating the immune defense of the host, which consequently escalates the development of secondary bacterial infection, most often caused by *Streptococcus pneumoniae*. Understanding the mechanisms leading to pathological disorders caused by bacterial superinfection after the previous viral infection is important for the development of more effective means of prevention; for example, by vaccination or through therapy using antiviral drugs targeted at critical viral proteins.

## 1. Introduction

Lower respiratory tract infections annually cause millions of human deaths worldwide [1]. A substantial portion of these deaths is attributable to seasonal influenza virus infections, due to the constant emergence of new variants of influenza A viruses (IAV) and synergistic infections with other viruses or bacteria [2,3]. The development of severe disease is often associated with the ability of primary viral infection to alter the host immune response, resulting in the promotion of the secondary infection. Such an infection can cause a potentially lethal disease associated with the systemic inflammatory response of the body [4,5,6].

Secondary bacterial infections can be caused by several bacterial species, such as *Acinetobacter baumannii*, *Haemophilus influenzae*, *Klebsiella pneumoniae*, *Mycobacterium tuberculosis*, *Pseudomonas aeruginosa*, *Staphylococcus aureus*, *Streptococcus pneumoniae*, or *Streptococcus pyogenes* [6,7,8,9]. The most common is the bacterium *Streptococcus pneumoniae*, which can persist in the human nasopharynx in a dormant state from early childhood to adulthood [6]. The dormant form of *S. pneumoniae* is non-invasive but can be reactivated and cause invasive infection after influenza infection [6,10,11]. Both IAV and *S. pneumoniae* are considered to be among the most important pathogens of the respiratory tract. An example of their synergistic relationship is the Spanish flu pandemic in 1918. It has been shown that the high mortality of this pandemic was not caused by the virus itself, but was instead the result of co-infection by these two pathogens, which led to synergistic pathologic disorders with devastating impacts [12]. Analyses of influenza-associated bacterial infections in mouse models have revealed higher morbidity and mortality rates in comparison to infection with the individual pathogens [13,14]. In the last three years, influenza viruses have been replaced by the coronavirus SARS-CoV-2 as the prevailing respiratory infection agents. Published data have shown that SARS-CoV-2 is also capable of making patients vulnerable to secondary bacterial infection [15,16,17]. Therefore, exploring and identifying the underlying mechanisms and the role of influenza virus proteins in establishing consequent bacterial infection would provide a new perspective on the prevention through vaccination and treatment, not only by antibacterial drugs, but preferentially by drugs against viral infections [15,18,19].

## 2. Pathogenesis of Co-Infection by IAV and *Streptococcus pneumoniae*

### 2.1. Viral Influenza Infection

Infection with influenza A viruses occurs in the upper airway epithelium. In the human population, the influenza virus spreads by droplets, and the manifestation of airway infection may be asymptomatic or with only very mild symptoms of uncomplicated upper respiratory tract infection. However, IAVs can also trigger complicated disease associated with severe pneumonia, leading to multi-organ failure or the worsening of existing health conditions, especially in individuals with immunodeficiency or chronic lung or heart disease [20,21,22]. The initial sites of influenza A infection are the pseudostratified columnar cells of the respiratory epithelium in the trachea, nasal cavity, and sub-mucosal nodes, as well as the pneumocytes in pulmonary alveoli. The virus further spreads and infects surrounding epithelial as well as immune cells, such as macrophages, dendritic cells, and NK cells [23,24].

IAV infection activates the immune response in the respiratory tract by triggering the innate defense mechanisms. IAV must first overcome the physical barrier, consisting of mucosal surface fluid containing antimicrobial peptides, neutralizing secretory antibodies, IgA, mucus, and a protective layer of basal and ciliated epithelial cells. The integrity of the airway epithelium is also under the constant surveillance of leukocytes [25,26]. Homeostasis is maintained through the release of cytokines, chemokines, and growth factors. When IAV enters cells, a signal is transmitted along the interferon signaling pathway and the antiviral response is activated. After pattern-recognizing receptors (PRRs) recognize virus-specific nucleic acids, they activate the transcription of interferon (IFN) genes and the secretion of proinflammatory cytokines. Secreted type I IFNs trigger the expression of IFN-stimulated genes in infected cells, as well as in adjacent uninfected cells, which encode a variety of antiviral proteins, thus protecting them from viral invasion [27]. IFNs are major cytokines with antiviral, antiproliferative, and immunomodulatory functions against viral and bacterial infections and represent the first line of defense against IAV infection [28]. When the IFN response and leukocytes do not prevent IAV replication, the pro-inflammatory response and apoptosis are activated. The presence of the virus also triggers the adaptive immune response, which is mediated by T-cells and B-cells, and the production of IgM and IgG antibodies, which can interfere with different stages of the viral life cycle [25]. However, IAV proteins can inhibit the functions of IFN at all levels, helping viruses to avoid the antiviral response [27,29]. The impaired antiviral response and activity of type I and type II interferons as a result of IAV infection inhibit the function of neutrophils and macrophages, as well as the response of T-cells and NK cells [30,31,32,33,34].

Influenza viruses contain eight gene segments that, together, encode nine structural proteins and at least nine non-structural proteins with regulatory functions [35,36]. The most abundant envelope protein of the IAV is hemagglutinin (HA), which is incorporated into the lipid bilayer derived from the infected cell [37,38]. It is a glycoprotein responsible for the attachment of the virus to the host cell receptor—sialic acid—and the fusion of viral and endosomal membranes, enabling the viral genome to be released into the cytoplasm [23,39]. During uncoating, the matrix protein 2 (M2), with ion channel activity, plays an essential role [40]. The inner side of the virion is lined by matrix protein 1 (M1), which determines the shape of the viral particles and surrounds the eight segments of the ribonucleoprotein (RNP) complex comprising the nucleoprotein (NP) and negative-sense RNA. Each of the eight segments is bound by its own polymerase complex, composed of three proteins (PB1, PB2, and PA) responsible for the replication and transcription of the genome segments [41]. The viral replication and suppression of immune response against IAV infection are mediated mainly by the multi-functional non-structural protein 1 (NS1) [42,43]. Viral RNA is synthetized *de novo* in the nucleus and is exported to the cytoplasm by the nuclear export protein (NEP), also known as non-structural protein 2 (NS2), which is found freely in the mature virion [44,45]. Newly translated proteins are transported to the cell membrane, where virus particles are completed and released from the infected cell with help of the exo-sialidase activity of second immunodominant protein, neuraminidase (NA) [41]. Novel accessory proteins of IAV, including PB1-F2, PB1-N40, PA-X, PB2-S1, matrix protein M42, and NS3, have recently been discovered. These are products of alternative reading frames within IAV gene segments, and they provide a replication advantage to the virus or emerge as a consequence of IAV adaptation to a new host [23,24,39,46].

The pathogenesis of IAV infection is characterized by two phases. The first phase lasts approximately 1–3 days and determines the maximum level of viral titer and the extent of the associated inflammatory response. Depending on these two parameters, the second phase may result in the acquisition of control over the virus transmission or end up as a widespread disease associated with acute respiratory distress syndrome and death. The clinical course of the infection and the outcome of the pathogenetic processes of the viral infection are determined by both viral and host factors [22,47,48]. The viral factors influencing the course of infection include the proteins encoded by the IAV [49,50,51,52]. The most variable viral proteins are HA and NA. At present, 18 different HA sub-types and 11 NA sub-types are known, which together can create more than 130 IAV sub-type combinations [53]. This means that there is a possibility of more potentially dangerous variants emerging in the future. Here, we discuss the roles of particular IAV proteins in modulating viral infection, viral–bacterial co-infection, and immune response.

### 2.2. Bacterial Infection with Streptococcus pneumoniae

The human upper respiratory tract is a suitable environment for colonization by the various micro-organisms constituting the human microbiome [54,55,56]. A frequently occurring bacterial species in the upper respiratory tract is *S. pneumoniae,* which is a Gram-positive extracellular pathogen belonging to a group of approximately 100 different serotypes [57] depending on an important virulence factor—the polysaccharide capsule [58,59,60]. *S. pneumoniae* form a biofilm, a highly organized multi-cellular community of one or more bacterial types that produce an extracellular matrix and adhere to abiotic and biological surfaces such as the human nasopharynx. The main characteristic of this biofilm is high antibiotic resistance and the presence of a protective matrix on the surface of the biofilm, which enables the avoidance of immune surveillance by the host and helps to maintain the persistence and dissemination of bacteria in the host organism [11,61,62,63,64]. The nasopharyngeal environment, low temperatures (34 °C), and limited nutrient supply are essential for the formation of a bacterial biofilm [65,66]. Depending on the part of the human body where the bacterium enters, such as the spinal cord and brain, heart, middle ear, sinuses, or lungs, it can cause various diseases, including bacterial meningitis [67], endocarditis [68], otitis media [69], sinusitis [70,71], and pneumococcal pneumonia [59], respectively. The ability of the bacteria to enter the body is limited by the maturity of the immune system; therefore, young children, the elderly, and immunocompromised individuals are the most vulnerable groups [58]. The mechanisms responsible for the disease-related transition of bacterial colonization from an asymptomatic biofilm in the nasopharynx to an invasive phenotype are not yet known [63,72,73,74,75].

### 2.3. Co-Pathogenesis of IAV and Streptococcus pneumoniae

Infection with IAV paves the way for *S. pneumoniae* invasion by causing respiratory tract damage, manifested by impaired integrity of the epithelial barrier, inflammation, and elevated glucose availability [76]. Several very serious complications can occur in the lung, such as a decrease or loss of gas exchange function, oedema, or pleurisy [7,77], which appear mostly in immuno-compromised patients, individuals with underlying genetic conditions, children under the age of five, and elderly adults above the age of 65 years. Extensive tissue damage in the alveolar space also results in a loss of the repair ability [78,79,80,81]. Loss of healing response to lung damage and also the loss of basal epithelial cells required for airway epithelial regeneration (including alveolar epithelial cells type I and II) are associated with an increased number of cellular receptors for bacterial adhesion, the attachment of bacteria to the surface of these cells, and their apoptosis. Defective tissue integrity promotes the development of complications ranging from bacteremia to sepsis [76,82,83,84]. The clearance of bacteria is slowed by decreased mucociliary velocity [7,85] and by the disruption of immune mechanisms. The depletion of alveolar macrophages and neutrophils, along with the increased production of anti-inflammatory cytokines such as IL-10, TGF-β, type I interferons, and IFN-γ, creates suitable conditions for bacteria dispersing from the biofilm (see Figure 1) [54,61,63,72,76,86,87,88].

#### 2.3.1. Disruption of Innate Immunity and Inflammatory Response during IAV and Bacterial Co-Infection

The first line of defense against respiratory pathogens includes the alveolar macrophages, which represent more than 90% of the immune cells found in the bronchoalveolar lavage fluid of a healthy individual [7]. These immune cells are specifically targeted by IAV, causing their depletion. As the proliferation and differentiation of new alveolar macrophages takes up to two weeks, the lung tissue is not sufficiently protected for this period of time, thus establishing a niche for the development of pneumococcal superinfection [89,90,91,92,93].

A crucial step in the development of a secondary bacterial infection is the disruption of the innate immunity and the inflammatory response. At the time of primary viral infection, when the viral titer decreases—usually 7 days after IAV infection—the anti-inflammatory response is very strongly induced, mainly by IL-10 or TGF-β produced by macrophages [94,95,96,97].

At this stage, IAV induces inhibition of the Th17 pathway, which suppresses bacterial clearance through NADPH-oxidase-dependent phagocytosis [31,98]. Previous recognition of viral dsRNA by the toll-like receptors (TLRs) of remaining alveolar macrophages causes an inability of the TLRs to recognize foreign bacterial products [90,91,99,100] and also triggers the production of IFN-γ by T-cells, preventing the clearance of *S. pneumoniae* by neutrophils and alveolar macrophages [32,101,102]. This step ensures that homeostasis is established but also reduces the ability of the immune system to recognize pathogens and to effectively defend itself [7,103,104].

In addition to the apoptosis of macrophages during IAV infection, the apoptosis of neutrophils [105,106], human dendritic cells [107], and NK cells [108] has also been observed. NK cells and T-cells produce immunomodulatory cytokines and mediate the cytotoxic response to viral infection; however, imbalances in their function during influenza infection can lead to an excessive inflammatory response and lung tissue damage [80,109,110,111,112].

#### 2.3.2. Autophagy and Apoptosis Mediated by Influenza Infection

Influenza viruses are able to activate, as well as inhibit, the host-cell apoptotic process, depending on the phase of infection [113,114,115,116]. Apoptosis appears to be linked with autophagy during IAV infection [117,118,119,120], which is important for maintaining cellular homeostasis [121,122,123,124]. Interactions among several IAV proteins, such as PB1-F2, NA, HA, NS1, nucleoprotein (NP), matrix protein 1 (M1), and matrix protein 2 (M2), manipulate autophagy and apoptosis. Recent *in vivo* and *in vitro* studies have shown that autophagy is viral-strain-dependent [116,118,125,126,127,128].

IAV alters autophagy and apoptosis in favor of viral replication and the release of new viral particles. In the early phase of infection (see Figure 1), viral NS1 protein up-regulates the synthesis of HA and M2, thereby indirectly promoting the formation of autophagosomes [115,117,119,120,129,130,131,132]. Autophagosomes are transient double-membrane vesicles, which are usually fused by lysosomes in the process known as autophagosome maturation. During the terminal stage of IAV infection, M2 protein interfere with the formation of autophagolysosomes [129,133,134,135]. Accumulated viral antigens enwrapped in autophagosomes avoid recognition by the immune system and antiviral response. At this phase of IAV infection, M2–dependent inhibition of autophagy promotes apoptosis for effective replication [117,118,119,120,135] and, consequently, induces damage to lung epithelial cell as well as the production of anti-inflammatory cytokines. All of these processes triggered by IAV contribute to the development of secondary bacterial infection by *S. pneumoniae* [93,102,118,136].

## 3. Role of IAV Proteins in Viral and Bacterial Co-Infection

Many studies have shown that, during primary infection, the influenza virus proteins facilitate bacterial infection, colonization, and disease development by *S. pneumoniae* in individuals of all ages [6,86,137]. Different IAV proteins are involved in the progression of bacterial superinfection through various mechanisms (Figure 1). NA facilitates access to receptors and nutrients for *S. pneumoniae*, which promotes the development of bacterial infection [23,138,139]. NS1 and PB1-F2 affect the regulation of the interferon response, the disruption of the inflammatory response, and/or the manipulation of the processes of autophagy and apoptosis [35,140,141]. Other IAV proteins have not yet been shown to play a direct role in the development of bacterial co-infection, but M1 [142], NP [143,144], and HA [119,129] facilitate the manipulation of apoptosis. M2 protein is required for the activation of inflammasomes in macrophages and dendritic cells [145], which indirectly helps in secondary bacterial infection development (Table 1) [35,146,147].

### 3.1. Characterization of PB1-F2 Protein

PB1-F2 is a small accessory non-structural protein found in some influenza A virus strains, encoded by the a + 1 alternative open reading frame of the PB1 segment [133]. The length of the protein varies according to the host specificity. The original avian PB1-F2 is composed of 87–90 amino acids [187], in contrast to 11 amino acids in human and swine isolates [51]. Zoonotic IAV strains represent a reservoir of full-length PB1-F2 sequences and were introduced into the human population during the pandemics of IAV strains H1N1 (1918), H2N2 (1957), and H3N2 (1968), characterized by an increased incidence of secondary bacterial infections [188]. Full-length PB1-F2 is a phosphoprotein with two helical domains, a C-terminus formed by an extended α-helix, and an N-terminus formed by two short α-helices, connected by a flexible and unstructured hinge region [189]. The PB1-F2 molecule can form oligomeric structures and membrane pores in the lipid bilayer [190,191]. The protein is mainly localized in mitochondria, but it can also be present in the nucleus and cytoplasm of infected cells [133,192,193]. Several virulence-associated amino acid residues and motifs have been identified within PB1-F2. These motifs gave PB1-F2 both intracellular and extracellular functions, such as the strain-specific regulation of polymerase activity [194] or the exacerbation of viral pathogenesis in animal models, direct or indirect induction of apoptosis, and the modulation of innate immune responses. To a large extent, PB1-F2 also participates in the activation of neutrophils, alveolar macrophages, and dendritic cells and in their recruitment into the airways [146,156,195]. The effect of PB1-F2 seems to be cell-type-dependent. With respect to these properties, PB1-F2 protein is able to influence the course of infection and enhance not only viral infection [51,146,147,195,196] but also subsequent secondary bacterial infections on several levels.

#### 3.1.1. Apoptosis and Cytotoxicity Mediated by PB1-F2

PB1-F2 itself increases IAV virulence by causing the cell death of the alveolar macrophages located in the lungs [197]. Their apoptosis leads to decreased antigen presentation, reduced initial viral clearance, and disrupted communication between alveolar macrophages and T-helper CD4+ cells. Consequently, the downstream effector functions of CD4+ cells, including the activation of cytotoxic T-lymphocytes (CTL), the production of antibodies, and the inflammatory response, are impaired [198]. Just one amino acid mutation at position 66 (N66S) can inhibit inflammasome activation and the IFN response induction, and increase the virulence of the virus [155]. PB1-F2 protein localized in mitochondria is able to promote apoptosis through mechanisms mediated by the mitochondrial targeting sequence at PB1-F2 C-terminus [199,200]. PB1-F2 localized in the outer mitochondrial membrane interacts with two mitochondrial membrane proteins involved in the formation of mitochondrial permeability transition pores, the voltage-dependent anion channel 1 (VDAC-1) and adenine nucleotide translocator 3 (ANT3). Their interaction causes the permeabilization of the mitochondrial outer membrane (MOMP), allowing cytochrome *c* efflux and resulting in apoptosis [150,201]. Moreover, PB1-F2 can form amyloid fibers and β-amyloid pore structures, leading to the permeabilization of cellular membranes and, subsequently, cell death [190]. The reduction of the pro-inflammatory response and promotion of apoptosis increase the frequency and severity of secondary bacterial infection *in vivo* [202].

The PB1-F2 protein of some IAV strains contains a cytotoxic sequence or “cytotoxic motif”, consisting of three amino acid residues (I68, L69 and V70) at the C-terminus, which can trigger cytotoxic death in immune and epithelial cells in lungs. This motif enhances immunopathological processes in lungs and accelerates the development of secondary bacterial infection [51,152].

#### 3.1.2. PB1-F2 Pathogenic Markers Enhancing Secondary Bacterial Infection

At present, five specific amino acid markers found in the C-terminal region of PB1-F2 are recognized, which are associated with the induction of the host inflammatory response [154], as well as with complications after secondary bacterial infection [203]. The first marker, amino acid exchange at position 66 in the PB1-F2 sequence (N66S), correlates with pathogenicity, increased virulence [155], and early IFN response inhibition [148]. IFN suppression results in the increased susceptibility of the host to secondary bacterial infection. The other four markers in the C-terminal region—specifically L62, R75, R79, and L82—represent the PB1-F2 “proinflammatory domain” or “inflammatory motif” linked with significantly higher morbidity and mortality in mouse models of all pandemic IAVs from 1918, 1957, and 1968 [51,153]. These “inflammatory residues” enhance the development of secondary bacterial pneumonia, as they are responsible for increased lung destruction and significant pulmonary inflammation by inflammatory cells and proinflammatory cytokines present in the lungs. Interestingly, PB1-F2 protein without changes in these locations (i.e., motifs P62, H75, Q79, and S82) is referred to as “noninflammatory protein” and possesses antibacterial activity [153].

Another function of PB1-F2 that influences the inflammatory response is its ability to regulate the NLRP3-inflammasome in humans and mice, which consequently induces the secretion of the pyrogenic interleukin IL-1β [156,204,205]. NLRP3-inflammasome is a cytoplasmic multi-protein complex that is activated upon IAV infection. This complex mediates proteolysis of interleukins from the IL-1 family to their mature and fully active form [206]. After IAV is recognized by NOD-like receptors (NLRs), the NLRP3-inflammasome complex is up-regulated and, subsequently, mature IL-1β is secreted [157,207]. PB1-F2 protein has been associated with the activation of the NLRP3-inflammasome through various mechanisms, such as the induction of apoptosis, the inhibition of IFN I activation, the acidification of phagolysosomes, mitochondrial disruption, the activation of reactive oxygen species (ROS), and the formation of aggregates of the PB1-F2 C-terminal region [156,157,204,205,207,208]. At the same time, PB1-F2 can also inhibit the activation of the NLRP3-inflammasome by decreasing the mitochondrial membrane potential. PB1-F2 translocates into mitochondrial inner membrane space by the major outer mitochondrial membrane channel (Tom40). Accumulated PB1-F2 protein decreases the membrane potential and accelerates mitochondrial fragmentation. The resulting pathway appears to depend on the secondary structure of the PB1-F2 spliceosomes [158,205,209].

Overall, PB1-F2 is a critical virulence factor; its properties and functions differ according to its amino acid sequence, strain specificity, cell type, and host specificity [210]. Although the exact mechanism by which this protein influences the virulence or immunopathogenesis is not yet fully understood, it is clear that PB1-F2 is involved in the enhancement of pathological processes during IAV infection, which may lead to the development of secondary bacterial infection [153,156,202].

### 3.2. Characterization of Hemagglutinin

Hemagglutinin (HA) is the major surface glycoprotein of influenza virus, which is essential for the onset of viral infection. It is encoded by the fourth segment of IAV and, as one of the virulence factors, is the main target of the immune response [41,211,212,213]. The synthesis of HA takes place on the ribosomes of the rough endoplasmic reticulum as a precursor molecule HA0, which is post-translationally proteolytically cleaved into two glycopeptides, HA1 and HA2, which are linked by disulfide bonds [38,214]. Depending on the nature of the cleavage site, HA0 can be cleaved intracellularly or extracellularly. Multi-basic cleavage sites are cleaved intracellularly with subtilisin-like cellular proteases (furin, PC6), and monobasic cleavage sites are cleaved extracellularly with trypsin-like serine proteases (tryptase Clara, HAT-protease, TMPRSS2, and plasmin) [215,216,217]. HA forms on the viral surface homotrimers. The globular domain of HA trimer (formed by HA1) ensures initial contact with the target cell through binding to sialic acid receptors present on the cell surface. The stem domain, created mainly by HA2 [218], mediates the viral–endosomal membrane fusion after low-pH-induced HA structural rearrangement and plays an essential role in the release of influenza virus genome into the cytoplasm [219,220].

#### 3.2.1. Changes of HA Cleavage during Viral and Bacterial Co-Infection

It has been shown that some bacteria influence the replication of the influenza virus and pathogenesis during secondary bacterial infection by promoting HA0 cleavage to its fusion-active form. This happens through the secretion of bacterial proteases or activation of cellular proteases [221]. The direct effect of proteases on HA cleavage and exacerbation of infection has been observed during co-infection with bacterial strains such as *Staphylococcus aureus* [222], *Aerococcus viridans* [223], *Streptomyces griseus* [224], and *Stenotrophomonas maltophilia* [225]. However, Callan et al. have described a reduction in viral infectivity after HA cleavage at an alternative amino acid sequence of the cleavage site with *Pseudomonas aeruginosa* protease [226]. Some bacteria in the respiratory tract produce proteins known as streptokinases and staphylokinases, which are capable of forming complexes that convert host enzymes into their active forms (e.g., plasmin, kallikrein, thrombin) and, thus, may indirectly increase the cleavage of the HA glycoprotein and spread of viral infection [223,227,228,229]. However, none of the aforementioned mechanisms are used during secondary bacterial infection with *S. pneumoniae*. As *S. pneumoniae* is able to bind plasminogen and host-derived activator onto its surface [230,231,232] and transport both proteins to the site of viral infection, it can possibly facilitate and increase the cleavage of viral HA into its active form [233]. In the case of viral–bacterial co-infection, the interaction between HA and bacteria leads to the promotion of viral infection. Viral HA, expressed on the surface of infected pneumocytes or on free extracellular virions, can bind to ligands on the polysaccharide capsule of *Staphylococcus aureus* [234], *Streptococcus pyogenes* [235], and *Streptococcus agalactiae* [236] and ameliorate bacterial internalization.

#### 3.2.2. The Role of Hemagglutinin in the Autophagy

HA of IAV is one of the proteins involved in the regulation of autophagy. Wang et al. have observed autophagy induced by HA binding to heat shock protein 90AA1 (HSP90AA1) present on the cell surface [159]. This strategy is exploited by IAV to prolong the time during which the virus replicates [118]. The process of exacerbation of the disease due to pneumonia is multi-factorial and dependent on viral, bacterial, and host factors [82]. The ways in which the other HA properties, such as HA tropism and affinity to sialic acid, extent of HA glycosylation, and the optimum pH for HA structural rearrangement into its fusion-active conformation, influence the development of bacterial superinfection are poorly understood at present [160,237].

### 3.3. Characterization of Neuraminidase

Neuraminidase (NA) and hemagglutinin (HA) are the major surface proteins of influenza viruses. NA is a tetramer comprising four identical monomers. Each NA monomer consists of four structural domains, namely, the catalytic head, the stem, the transmembrane region, and the cytoplasmic tail [23,238]. Four catalytic head domains create the enzymatic site with exo-sialidase activity. The stalk domain contributes to the stability of the NA tetramer. The structure and length of the stem differ among viral strains. Its length, in particular, affects the ability of the virus to replicate [238,239,240]. The transmembrane domain anchors the NA tetramer to the viral envelope, and its signals are necessary for translocation from the endoplasmic reticulum. The fourth domain—the cytoplasmic tail—is thought to interact with matrix M1 viral protein and affects virion morphology and virulence [238,241,242,243]. Neuraminidase is an enzyme that cleaves the terminal α-glycosidic bond between N-acetyl-neuraminic acid (sialic acid) and the carbohydrate residues of glycopeptides or glycolipids on the cell surface. The NA of influenza viruses functions at multiple levels during IAV infection. NA activity enables the release of *de novo* synthesized budding viral particles from the infected cell surface and prevents virion aggregation. This cleavage mechanism of NA prevents not only the aggregation of emerging virions but also the rebinding of these virions to the dying host cell [23,238,244]. Viral NA also cleaves neuraminic acid residues from airway mucin, allowing the movement of the virus into target cells [23,238,245]. Another function is its role in enhancing HA-mediated membrane fusion [238,246].

#### 3.3.1. The Role of the Viral and Bacterial Neuraminidases in Co-Infection

Many bacterial receptors are coated with carbohydrates covered by sialic acids. Therefore, most bacterial strains synthesize their own bacterial neuraminidases, which cleave sialic acids on the host cell surfaces and allow for their adhesion. *S. pneumoniae* expresses three types of neuraminidases: NanA, NanB, and NanC [247,248]. The most expressed and active is NanA, with a conserved catalytic site present in all strains. Although viral and pneumococcal neuraminidases have different quaternary structures (bacterial NanA is a monomer), their active sites are similar but have different substrate specificities [249,250]. The epithelium of the upper part of the respiratory tract is densely coated with α2,6-linked sialic acids, while α2,3-linked sialic acids prevail in the lower respiratory tract. Microbial NanA and viral NA can cleave sialic acids bound by both types of glycosidic bonds [249]. *S. pneumoniae* NanA facilitates colonization of the respiratory tract by cleaving sialic acid from mucin and reducing the viscosity of the mucus. Influenza infection increases the production of mucus, which serves as a source of nutrients for *S. pneumoniae* [6]. Therefore, after previous infection with influenza viruses, pneumococci can adhere to the pulmonary epithelium and invade host cells to a greater extent. This has been described especially for more virulent strains of the H3N2 sub-type, revealing relatively high viral neuraminidase activity. Several sources have suggested viral–bacterial neuraminidase interactions [18,77,251,252].

The NA activity of IAV alters the glycosylation of the host cell surface in the airway epithelium, thus affecting local and systemic immune responses and enhancing the development of bacterial infection [138,139]. Neuraminidases of IAV and *S. p**neumoniae* can desialylate surface glycans on lung epithelial cells and expose sub-terminal galactosyl groups as ligands for soluble β-galactoside-binding proteins (i.e., galectins).

#### 3.3.2. Cooperation of NA and Galectins during Co-Infection

Galectins play a key regulatory role in cell development and immune homeostasis. They are involved in cell activation, differentiation, and signaling. Galectins also serve as immune recognition receptors and effector factors in leukocyte recruitment and apoptosis and mediate host–pathogen interactions [253,254,255,256,257,258]. Yang et al. have recently identified three galectins with anti-influenza effects—Gal-1, Gal-2, and Gal-3—released after IAV infection from various cells in the respiratory tract [258]. Current studies have confirmed that Gal-1, Gal-3, and Gal-9 alleviate the overall course of influenza infection by blocking viral binding and strengthening cell immune responses [162,258,259,260,261,262].

It has been found that primary IAV infection and subsequent *S. pneumoniae* infection increase galectin expression and release, including Gal-1 and Gal-3, which bind strongly to glycans on the surface of both pathogens. Subsequently, galectins, including galectin-3, bound on the surface of *S. pneumoniae* interact with ligands and mediate the adhesion of *S. pneumoniae* to epithelial cells and, as has been shown in both *in vitro* and *in vivo* models, support *S. pneumoniae* invasion during co-infection [162,263,264,265,266,267]. This interplay between galectins and viral NA may determine the intensity of pneumococcal infection.

#### 3.3.3. Impact of the Viral NA Activity on the Innate Immunity

Neuraminidases of the influenza virus, as well as from *S. pneumoniae*, are some of the few reported pathogen-encoded molecules that directly activate an important cytokine, transforming growth factor β (TGF-β). TGF-β is an important mediator of interactions between the infectious pathogen and its host and is constantly expressed in the body in the form of mRNA or present as a latent protein. Thus, it rapidly becomes accessible in the event of a primary infection. The latent TGF-β consists of an N-terminal latency-associated peptide (LAP), which is non-covalently linked to a C-terminal mature TGF-β1 molecule. The release of mature TGF-β from LAP by NA activates the TGF-β, which binds to cellular receptors and induces a biological response [163,268,269]. In the respiratory tract, the expression of the TGF-β1 isoform controls the differentiation, proliferation, and activation status of leukocytes. TGF-β is an immunosuppressive molecule, and influenza infection potentiates its immunosuppressive ability. Elevated TGF-β levels can induce apoptosis of both immune cells and lung epithelial cells, confirming the potential role of NA in the pathogenesis not only of viral infection, but also of bacterial co-infection [163,270,271,272]. TGF-β, a regulator of the adaptive immune response, reduces the number of activated cytotoxic T-cells and induces the production of IL-10 by regulatory T-cells, thus facilitating lung colonization by *S. pneumoniae* [164,273]. Although TGF-β is not an effector cytokine that can mediate bacterial clearance, Roberts et al. have shown that hosts with allergic airway disease, which induces TGF-β production, were protected against severe influenza and bacterial co-infection [273]. On the contrary, Li et al. have demonstrated that TGF-β directly activated by influenza virus NA promotes the expression of cellular adhesins, leading to decreased bacterial clearance and increased colonization. Therefore, the exact role of NA-activated TGF-β in co-infection remains unclear [164].

### 3.4. Characterization of NS1

Non-structural protein 1 (NS1) has many functions and is known as a major viral antagonist of the interferon response [274,275,276]. Thus, NS1 represents a very important virulence factor affecting the pathogenesis of influenza disease, as well as viral and bacterial co-infection [35,141,277].

NS1 is a product of alternative splicing of the NS gene, occurring as a homodimer. The monomer is composed of two functional domains, the N-terminal RNA-binding domain (RBD) and the C-terminal effector domain, which are joined by a short inter-domain linker region [35,141,278]. Each domain interacts with different cellular factors. The homodimer of NS1 can bind to various species of RNA—most importantly, to double-stranded RNA (dsRNA) by RBD at the N-terminus. This binding property influences multiple functions of cellular proteins critical in the development of secondary bacterial infection, consequently affecting the onset, course, and severity of the bacterial co-infection [35,140,277]. The most significant is the influence of NS1 on the type I IFN response as an antagonist of type I IFN signaling. NS1 also interacts with other IAV and host proteins, controlling the autophagy and apoptosis of lung epithelial cells [279,280,281,282].

#### 3.4.1. NS1 Interaction with Interferon Signaling Pathways Enhances the Development of Secondary Bacterial Infection

After the IAV invades the host, it is recognized by a cellular pathogen sensor called retinoic acid-inducible gene-I (RIG-I). Subsequently, the RIG-I-mediated signaling pathway induces the expression of interferons [283]. However, NS1 is capable of inhibiting this pathway in several ways: first, by the direct interaction with E3 ubiquitin ligases TRIM25 [166] and Riplet [167], which prevents the activation of the RIG-I receptor; second, indirectly by interaction with host factors through its effector domain at the C-terminus, which leads to the inhibition of IRF3 phosphorylation [141,174] and subsequent inhibition of the type I IFN response mediated by IRF-3, as well as interferon-stimulated genes (ISGs) [141,175,284]. These steps result in the prevention of virus detection by host cells [166,276,285,286,287]. Next, NS1 can bind its RBD domain to RIG-I and several other proteins with dsRNA-binding ability and block their activation—for example, dsRNA-dependent serine/threonine–protein kinase R (PKR) [168,276,288].

PKR is a crucial protein responsible for the host antiviral response. It activates nuclear factor κB (NF-κB), thus contributing to type I IFN response [168,276]. During IAV infection, NF-κB regulates the expression of many cytokine and chemokine genes, including the antiviral cytokine IFNβ [289]. PKR activates IKκB, part of the IKK kinase complex, which phosphorylates the NF-κB inhibitor IκB, resulting in the activation of NF-κβ [290,291]. PKR also significantly slows down viral protein synthesis through phosphorylation of eukaryotic initiation factor-2 α-subunit protein (eIF-2α). Binding with NS1 prevents its activation, which blocks subsequent antiviral responses [168,276,292]. IAV viruses with defective or deleted NS1 (delNS1 mutants) are unable to block PKR activation [169]. They can, however, replicate in the absence of PKR, which proves the role of NS1 in counteracting the PKR-mediated antiviral response [170]. Furthermore, the RBD of NS1 can also compete with the RNA-binding capacity of oligoadenylate synthase (OAS), which is able to cleave viral RNA by the activation of RNase L. Such degraded viral RNA elements can be recognized by RIG-I receptors that activate IFN production [141,172,274]. All of these steps and functions weaken the immune response and prevent the detection of the virus in the host cells, suggesting the important role of NS1 in enhancing the development of secondary bacterial infection [33].

#### 3.4.2. NS1 Motif Directly Involved in Co-Infection with *S. pneumoniae*

The structure and function of NS1 differ among types of influenza viruses, and even among sub-types, and may be used to predict the potential of a particular strain to enhance the development of bacterial co-infection [29,203]. It has been reported that influenza infection with A/Puerto Rico/8/34-H1N1 (PR8) virus impaired IFN-β production, which led to increased susceptibility to secondary bacterial infection [282]. At 7 dpi, when the organism is most susceptible to the development of secondary bacterial infection [6], minimal IFN-β production [43,282,285] and extensive inflammatory response with insufficient bacterial clearance were observed [9,31,293]. This environment provides suitable conditions allowing for the rapid spread of bacteria, thus causing tissue damage. In 2019, Shepardson et al. have identified HA and NS1 of the influenza virus as individual regulators of secondary bacterial infection severity and described a motif binding PDZ (PDZ-bm) present at the C-terminus of the NS1 protein of the PR8. To unveil its function, the PDZ-bm domain was deleted from the NS1 protein of PR8, and the resulting virus was not capable of lowering the expression of IFN-β. Thus, they identified the PDZ-bm sequence, which is directly involved in controlling the susceptibility to secondary bacterial infection through the regulation of the IFN-β response [282]. As there are significant differences among influenza virus strains in this region, it was suggested that NS1 proteins from different viruses have different impacts on the host susceptibility to secondary bacterial infection.

#### 3.4.3. NS1 Manipulates Apoptosis

NS1 exhibits a complex role in the modulation of apoptosis, depending on various factors including the virus strain and the cell type. It is well-known that NS1 can, for example, directly bind to the linker region of PKR by its effector domain and block PKR-mediated apoptosis by preventing its conformational change and autophosphorylation of PKR. NS1 can also activate the host cell phosphatidylinositol 3-kinase (PI3K) pathway and, in this way, impair apoptosis [276,277]. The NS1 of H3N2 and H5N1 sub-types interact with heatshock protein Hsp90 and mediate the apoptosis of lymphocytes through the caspase cascade [181]. However, the interactions between NS1 and apoptotic pathways are still not well-understood and need to be further described in the future.

## 4. Conclusions

Influenza A virus infections in humans cause an acute respiratory disease with a spectrum of clinical symptoms. In many patients, the disease can be life-threatening and associated with various complications, depending on the comorbidities of the infected individual. The most frequently occurring complications are linked with bacterial co-infections or secondary superinfections, which enhance the pathological processes triggered by the IAV infection.

At present, many unanswered questions remain regarding the role of the host and its defense mechanisms in infection control, as well as about the effect of bacterial co-infection on the course of influenza infection and the host immune response. Additionally, many questions arise concerning the role of the primary viral infection and the function of viral proteins in this process [2]. Influenza infection damages lung tissue and promotes bacterial colonization through the alteration of the cytokine response. After the virus disrupts the epithelial barrier and induces the production of cytokines, thus reducing the number of Th17 cells in the lungs while also disrupting macrophage function and the production of suppressive cytokines released by regulatory T-cells, the primary influenza infection can lead to the development of a bacterial superinfection. Thus, the innate immune response does not function properly after being affected by viral proteins such as HA, NA, PB1-F2, and NS1. At least these IAV proteins are responsible for the disruption of the innate immunity and inflammatory response during viral and bacterial co-infection, as well as for the inability to trigger the antiviral response of the infected host. However, the synergy of the pathological impact of influenza–bacterial co-infections is a complex process, where not only viral but also bacterial and host factors play an important role. While indirect evidence of the roles of the various viral proteins in the pathogenesis of secondary bacterial infection has been supported by animal studies, exact direct clinical evidence in humans is, in general, lacking. Not only has such synergy has been observed during IAV pandemics or epidemics, but there are now data regarding similar complications in humans during SARS-CoV-2 infections. Understanding the mechanisms causing the devastating pneumonia through viral–bacterial co-infections can bring insight into the finding of targets for effective antiviral drugs or may help to improve prevention efforts through effective vaccines, consequently preventing the severe course of potentially dangerous respiratory infections.

## Figures and Tables

**Figure 1 viruses-14-01064-f001:**
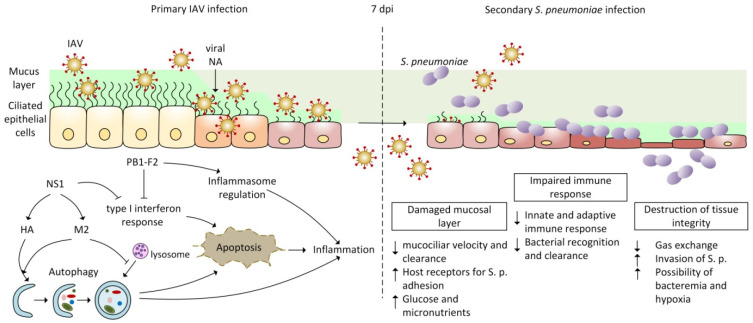
Illustration of influenza virus protein functions in the enhancement of *S. pneumoniae* lung infection. The mucosal layer in the lungs is damaged due to sialidase activity of viral NA, and ciliated epithelial cells in lungs, together with immune cells, are infected with IAV. During infection, IAV proteins PB1-F2, NS1, HA, NA, and M2 cause impairment of the immune response, the apoptosis and inflammation, leading to the destruction of tissue integrity. All of these viral protein functions can lead to enhanced susceptibility to secondary *S. pneumoniae* infection observed approximately 7 days after viral infection (7 dpi).

**Table 1 viruses-14-01064-t001:** Function of IAV proteins predisposing host to pneumococcal co-infection.

IAV Protein	Function	Functional Domain	References
PB1-F2	Inhibition of IFN response	N66S mutation	[148,149]
	Apoptosis of epithelial and immune cells	-	[150,151]
	Cytotoxic death of epithelial and immune cells	Cytotoxic motif	[152]
	Induction of rapid inflammatory response	Inflammatory motif	[153,154]
	Regulation of NLRP3 inflammasome activity	-	[155,156,157,158]
HA	Regulation of autophagosome formation	-	[119,127,159,160]
NA	**Creation of environment for bacterial entrance**		
	Alteration of glycosylation on cell surface	Catalytic domain	[138,139]
	Desialylation of surface glycans		[161,162]
	**Affection of innate immunity**		
	Direct activation of TGF-β	Catalytic domain	[163,164,165]
NS1	**Inhibition of IFN response in several ways**		
	Blocking of RIG-I activation	RNA-binding domain	[166,167,168]
	Blocking of PKR activation		[168,169,170]
	Blocking OAS function		[171,172]
	Interaction with host factors	Effector domain PDZ-binding motif	[173,174,175,176]
	**Manipulation of apoptosis in several ways**		
	Binding to PKR linker domain	Effector domain	[120,132]
	Activation of PI3K pathway	SH3-binding motif aa 164–167	[119,177,178,179]
	Interaction with Hsp90	-	[180,181]
	Inhibition of p53	aa 144–188	[182,183,184]
M2	Induction of autophagosome formation	-	[119,124,129]
	Inhibition of lysosomal degradation of autophagosomes	-	[129,185,186]
NP	Induction of autophagosome formation	-	[128,129]

(-) Multiple mechanisms or not defined.

## Data Availability

Data is contained within the article.

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
