# Peer review of "The Contribution of Viral Proteins to the Synergy of Influenza and Bacterial Co-Infection"

_viruses, 2022, doi:10.3390/v14051064_

Round 1

Reviewer 1 Report

The peer-reviewed manuscript of Miriam Mikušová et al. "The contribution of viral proteins to the synergy of influenza and bacterial co-infection" focuses on the role of influenza virus proteins (PB1 F2, NA, HA, M2, and NS1), which are key players in modulation of viral infection as well as immune response of host, and also in creating conditions for result in bacterial co-infection. In my opinion, this is an original and interesting analysis of data on the mechanisms of viral pneumonia development. It draws attention to the possible existence of a synergy of viral, bacterial and host factors in the development of a severe viral pneumonia. The review is well structured, it analyzes mainly data for the last 15 years.

Major point

The fragment about cooperation between viral NA and galectins (3.3.2) is difficult to read, it is possibly that this fragment contains contradictions. It needs careful editing and correction of technical errors.

Minor point

Figure 1. In the picture at the top: I.week and II.week. What is the meaning of these points? Is it necessary to specify "7DPI"? Specify whether the NA at the top of the left figure is host or viral?

Line 343: Reference [293] is out of order in the text.

Table 1 Subheadings are not uniform.

Author Response

Institute of Virology, Biomedical Research Center, Slovak Academy of Sciences,

Dúbravská cesta 9, 845 05 Bratislava, Slovak Republic

Topic:

Manuscript ID: viruses-1693286

Type of manuscript: Review

Title: The contribution of viral proteins to the synergy of influenza and bacterial co-infection

Authors: Miriam Mikušová, Karolína Tomčíková, Katarína Briestenská, František Kostolanský, Eva Varečková *

Dear editor,

journal “Viruses”

Thank you for your letter and for all the recommendations of changes in our manuscript. We implemented all requirements of referees and we corrected the Table 1 according to your instructions. 

We send you the text of our review article after the minor revision and hope the article in this form could be acceptable for the publication in the journal “Viruses”.

Sincerely,

Dr. Varečková Eva

corresponding author

Leader of the project

viruevar@savba.sk

Tel: 421-2-59302 427

Bratislava, 12th  May 2022

Cover letter: reply to the requirement of the minor revision

Reviewer 2 Report

Mikusova have provided a well documented and thorough evaluation of the complex interplay between influenza A virus and bacterial proteins that can promote secondary bacterial infection. There are two minor critiques: 

While there is indirect evidence of the role of the various viral protein in the pathogenesis of secondary bacterial infection that in some cases is supported by animal studies, direct clinical evidence is in general lacking. This fact should be so stated in the conclusions. 

Table 1 as reproduced is very confusing as to the correlation between proteins in the 1st column and items in successive columns. The authors need to ensure that the table has been carefully edited. 

While the English usage is reasonable, there are many minor errors in English language syntax and sentence structure throughout the paper that need to be carefully reviewed and corrected.  
